# Balancing conservation and human access to nature: the impact of a constructed causeway on water levels and sedimentation, North Bull Island, Ireland

Tidal dynamics; coastal management; salt marsh; sea-level rise; climate adaptation

**Corresponding author:**
I. Möller;
Email: moelleri@tcd.ie

## I. Möller [ID] and K. O'Leary

Department of Geography, School of Natural Sciences, Trinity College Dublin, Dublin, Ireland

## Abstract

With coastal populations rising at three times the global average, sustainable ways of safeguarding human needs around access and use of the coast alongside lasting ecosystem health of coastal environments must be developed. At the same time, human populations are facing the challenge of managing coastal access on the back of a legacy of human interventions that have already altered – and have often had unintended or unforeseen impacts on – the coastal system and its functioning.

We chart the history of the evolution of North Bull Island in Dublin Bay as an example of major unforeseen sedimentation in a coastal estuarine bay following the construction of river mouth training walls. We investigate the impact of a constructed causeway on the evolved 'naturescape' by comparing accretion and elevation change on the mid-marsh either side of the access road over a 32-month period (autumn 2021 to summer 2024) and measuring water levels either side of the causeway on six spring tides on consecutive days characterised by varying meteorological conditions in early September 2023. The results allow us to consider the potential implications a lack of physical connectivity may cause for the future of the two artificially separated back-barrier lagoon environments.

## Impact statement

Coastal populations are rising at three times the global average. This population increase is particularly marked in urban areas, including Dublin Bay, Ireland. It has led to interventions that have changed the way in which tidal and wave processes alter coastal environments. Waves and tides are particularly important in determining where erosion and deposition occur and thus where coastal ecosystems can form and persist. At the same time as our actions have altered coastal environments, we have become more aware than ever of the many benefits coastal environments provide human society with, such as biodiversity, carbon stores, and buffers against extreme storms. It is imperative that we understand what impact our past interventions are having on the functioning of critical coastal habitats so that we can take actions to ensure that the benefits provided are maintained into the future.

Here, we focus specifically on North Bull Island in Dublin Bay. The island's formation was affected by the construction of two walls on either side of the River Liffey. To the lee of the wall in the northern part of the Bay, sediment accumulated to form the island around 200 years ago. We provide the first empirical evidence that tidal water levels rise and fall very differently in the lagoons north and south of the constructed causeway. As the two parts of the lagoon have become disconnected through the causeway, we also show that this altered flood and ebb flow has impacted sediment accumulation and surface elevation of the tidal marshes on either side of the causeway; accelerated sea level rise will bring about a different response in either lagoon. Our results critically inform future actions needed to safeguard the habitats that occupy this coastal space and that currently act as important carbon stores, biodiversity hubs and flood mitigation features.



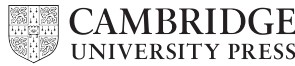

## Introduction

Coastal environments provide a particularly challenging setting in which human needs and natural dynamics must be managed to accommodate a constantly changing context. Urbanisation, the human use of the coast, rising sea levels, extreme storm events and river flooding are just some of the drivers of human interventions on dynamic coastal fringes (Hinkel et al., 2018).

Historically, human interventions at the coast have often taken the form of hard engineered structures, designed to break, divert, or reflect the sea's energy (currents generated by waves, tides, or storm surges), but history has shown us that such interventions are often at odds with the sustainable functioning of the coastal process environment, and frequently unanticipated

(negative) consequences have resulted from human actions designed to lead to specific (positive) outcomes (Temmerman et al., 2013). With the recognition of ecosystem services (Costanza et al., 1997) emerged the concept of nature-based solutions, that is, solutions that draw on nature's ability to provide services that solve particular challenges, such as the need for coastal protection or carbon sequestration (Morris et al., 2018). In the coastal context, and particularly on urban shores with a long history of human interventions, however, it is becoming increasingly obvious that the use of 'nature-based solutions' in an urban coastal context remains underexplored (Louarn et al., 2025). Further, it is becoming clear that 'nature-based solutions' are still fundamentally 'human solutions', requiring trade-offs between the needs of diverse human groups as well as the natural environment they depend on (Tozer et al., 2022).

In this context, it is critical that the focus moves away from framing decisions as either 'nature-based solutions' or 'hard engineering' and that solutions are sought that aim to balance the needs of human society as well as those of a dynamic coastal environment (albeit heavily human-modified), sustaining the health of critical ecosystems. Achieving sustainable interactions between humans and the coastal environment requires an understanding of the functioning of complex coastal systems in the context of our interventions and, ideally, as part of a landscape/seascape scale approach (Slaymaker et al., 2021).

Here, we focus on the case study of Dublin, Ireland, where, in the aftermath of the construction of two training walls designed to encourage the self-dredging of the River Liffey, sediment accumulated rapidly in the northern Dublin Bay around the 1800s (Jeffrey et al., 1977). Over the following decades, the sand shoal stabilised and evolved into a barrier island complex, consisting of an extensive beach, dune and back-barrier lagoon (see further detail below). Whether the island was a direct consequence of the training wall construction remains unproven, but it appears that the wall's diversion of river flow into the outer bay was able to create a more shallow environment conducive to the settling of fine silts and sands. Importantly, the beach and dune environments became rapidly used as areas of recreation (with the Royal Dublin Golf Club being granted permission to establish a course there in 1889), initially accessible by boat or on foot at low water and later by causeway. The back-barrier lagoon's tidal flats and salt marshes (alongside the beach and dune ecosystems) are now recognised as providing a multitude of ecosystem services (Dublin Bay Biosphere Partnership, 2022).

In this study, we aim to shed light on the impact of the access causeway on the present-day geomorphological functioning of the back-barrier lagoon and particularly the sedimentation and elevation changes that manifest on the salt marsh ecosystems, that is, we ask whether providing access to this space by way of a causeway (rather than alternative modes of access) has altered the environmental processes required to sustain one of the highly valued island ecosystems, the salt marsh. We do so by (a) establishing whether the salt marshes on both sides of the causeway are keeping pace, at least vertically, with rates of sea-level rise as recorded over recent decades and (b) investigating whether any differences in the depositional regimes on either side of the causeway may be due to differences in tidal inundation patterns on either side of the causeway.

## Field site

Dublin Bay, in the northern part of which North Bull Island is found, is characterised by extensive deposits of reworked, predominantly glacial, sediments, bound to the north by the rocky outcrop of the Howth peninsula (connected to the mainland through post-glacial sedimentation forming a tombolo) and, to the south, by Dalkey Hill (Mathew et al., 2019). The sands and silts that characterise the bay sediments date back to the last Celtic (British-Irish) Ice Sheet, which reached its maximum extent between 28,000 and 20,000 years before present. The shallow slopes of the coastal River Liffey, which enters Dublin Bay from the East, provided an extensive sheltered depositional environment. Post-glacial dynamics within the Bay have been characterised by the dynamic interactions between intertidal flats, braided estuarine channels, sandflats, dunes and salt marsh with early Neolithic settlers taking advantage of the fertile river floodplain (Smyth, 2014).

Given this geological/geomorphological context, it is perhaps not surprising that urban growth and the rise in the importance of Dublin as a commercial port ultimately required the city to address the problem of siltation of the main river channel. Consequently, the construction of the south and north 'Bull Walls' either side of the River Liffey in the 1730s (south) and 1820s (north), respectively, was designed to lead to greater 'self-scouring' of the riverbed through accelerated ebb tide flows (Kennedy, 1949).

The intervention was largely successful and is likely the reason why river sediment loads entering the bay beyond the training walls led to the growth of the already shallow intertidal sand flats north of the North Bull Wall, finally emerging to form small islands and ultimately coalescing to form the now ca. 5 km-long barrier island of North Bull (Figure 1).

As the island's beach, dune and back-barrier lagoon developed, their value as a recreational space increased, with access to the island via a wooden bridge connecting the north Bull Wall to the suburb of Clontarf. In 1889 the first, and in the 1920s the second, golf course was constructed on the island, followed, in the 1960s, by a causeway facilitating road access. Recent years have seen the island visited by up to 1.4 million visitors every year (Dublin City Council, 2020). The island's value as a provider of multiple ecosystem services beyond that of recreation has become increasingly recognised, not least through its location within the UNESCO Biosphere Reserve (Dublin Bay Biosphere Partnership, 2022). The vegetation on the salt marshes in the back-barrier lagoon, both north and south of the causeway, is predominantly that of a typical Atlantic salt meadow, composed of a mixture of *Aster tripolium, Puccinellia maritima, Atriplex portulacoides, Juncus maritimus* and *Festuca rubra*, as well as *Spartina* spp. (Cruz et al., 2019).

Sea level rise at the two main nearby tide gauges in Dublin and Howth, located within a 2 km distance to the south-west and north-east, has been estimated at 6.48 mm year$^{-1}$, well above the global average, over the period 1997–2016 (Shoari Nejad et al., 2022). However, little is known about how the division of the back-barrier lagoon by the causeway into two separated lagoons has been and will be affecting the future of the ecologically valuable back-barrier environments in this context.

## Methods

To address our key question as to whether providing access to the island by way of a causeway has altered the environmental processes required to sustain the salt marsh ecosystems and begin to better understand the process environment leading to potential longer-term differences in the back-barrier lagoon's response to a rising sea level, we conducted measurements of (a) surface elevation change, (b) accretion and (c) surface sediment characteristics on the

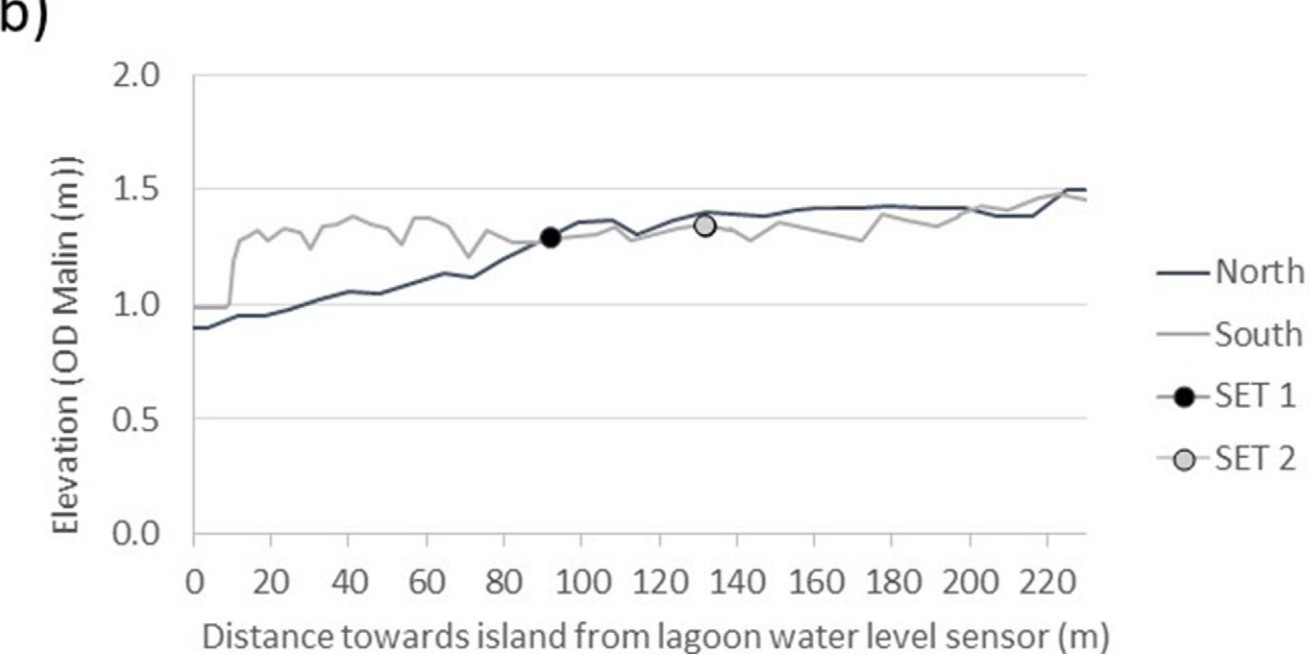

**Figure 1.** (a) location of Dublin on the east coast of Ireland (inset) and (b) the position of sedimentation elevation tables (SET) sites and water level sensors on North Bull Island, as well as the position of the Dublin Port tide gauge.

upper marsh, as well as (d) tidal inundation in the lagoons north and south of the causeway.

### Marsh surface elevation change

We deployed two Sedimentation-Elevation-Tables (SETs) on the upper, densely vegetated, salt marsh surfaces either side of the causeway at similar distance to the most landward dune-to-marsh transition (see Figure 1). The SET methodology is based on recording the distance between a horizontal datum (provided by a levelled metal bar placed on a fixed base station) several centimetres above the marsh surface and the marsh soil surface below. Instead of simply recording the change in the elevation of the surface to a datum inserted at the surface, however, the change in elevation is measured against a datum at least 2 m below the present marsh surface (i.e., the rod on which the horizontal bar sits is anchored in basal sediment that is fully compacted at depth below the marsh surface). Changes in the vertical distance between the soil surface and the reference bar thus represent the cumulative effect of surface sediment removal or addition plus any compaction of already deposited sediment relative to the deep base layer. At each SET station, measurements are taken along short transects (nine measurements at 5 cm intervals) along each of the four compass directions, yielding $9 \times 4 = 36$ readings per station (Callaway et al., 2013). Both stations were installed in the autumn of 2021, with the first (baseline) reading conducted on the 19 November 2021. Subsequent readings were taken on 10 August 2022, 23 August 2023 and 16 August 2024, thus capturing 33 months of elevation change.

### Marsh surface accretion

In addition to recording the change in the elevation of the marsh surface relative to the deep compacted sediment below the marsh surface, marsh surface accretion (i.e., addition of sediment to the marsh soil surface) was measured following the method described in Cahoon et al. (2000). We deployed a kaolinite clay layer of *ca* 2 mm thickness as a marker horizon on the marsh surface in the autumn of 2021 (when the SET stations were installed). The marker horizon was placed at a distance of approximately 30 cm outward from the SET measurement bar at each of the four SET bar locations. Cores were then extracted after several months to determine the depth of added surface sediment above the marker horizon. These measurements were taken at the same time at which elevation change was recorded using the SET technique. Three replicates of surface accretion were sampled per SET bar position, yielding 12 measurements per SET location (Figure 1).

### Surface sediment characteristics

The proportion of organic matter (e.g., resulting from plant litter and below-ground root production) relative to minerogenic matter (as delivered by aeolian or hydrodynamic processes) is a key determinant of the degree of compaction of salt marsh surface soils (whereby soils with a higher percentage of organic matter content compact more over time; see, e.g., Saintilan et al., 2022). Furthermore, the particle size distribution of surface sediments can provide clues as to the source and pathway of deposited sediments on salt marsh surfaces (e.g., De Groot et al., 2011). To determine the percentage of organic matter content and the particle size of surface sediments north and south of the causeway, we sampled 12 surface scrapes on 18 February 2022 (four months after the installation of the SET stations) near the two SET stations.

Organic matter, as a percentage dry weight of sampled sediment, was established via loss-on-ignition, in keeping with the methodology of Grey et al. (2021), whereby samples were returned to the laboratory immediately upon sampling, weighed and dried at 65°C for 48 h until no further weight loss occurred. Samples were allowed to cool in a desiccator prior to establishing their dry weight and moisture content (weight loss upon drying) and were then placed in a muffle furnace at 550°C for four hours. Once cooled, samples were re-weighed to establish the percentage weight of organic matter lost (loss-on-ignition).

Particle size of the surface sediments >63 μm was determined by dry (mechanical) sieving surface sediment scrapes collected at approximately the same locations as the organic matter samples, although only two samples each were collected in the 20 × 20 cm areas. Samples contained an average of 440 g sediment and were sieved through a mechanical sieve rack of sizes 355, 250, 180, 125, 90 and 63 μm for 8–10 min or until all aggregated sediment grains were fully separated and dislodged. The particle size analysis package Gradistat Version 9.1 (which runs in MS Excel) was used to compute basic particle size statistics. The package uses the Folk and Ward (1957) method to derive physical particle size descriptors.

### Water level measurements

Water levels were recorded over six tides at various parts of the spring-neap tidal cycle and under varying meteorological conditions in late August to early September 2023. An additional sixth tide was monitored in late September when meteorological conditions contrasted with those of the earlier tides. Records of water level were obtained using In Situ Rugged Troll 100 loggers, mounted either side of the causeway in the centre of the northern and southern lagoons and set to record pressure above the tidal flat surface at a frequency of 10 min. This allowed the establishment of tidal flooding/draining patterns for six tides for comparison with tidal stage measurements acquired by a tide gauge operated in Dublin Port (see also Shoari Nejad et al., 2022).

Water level sensor positions are shown in Figure 1. The elevations of both sensors were recorded using a Trimble RTK dGPS instrument with a vertical measurement accuracy of ±5 cm in September 2023. In addition, sensor elevations were levelled against a common datum (a metal pin inserted in concrete) on the causeway using a standard dumpy level to verify their elevations relative to one another. The sensor installed north of the causeway was at a height of 0.659 m OD Malin Head, while the sensor in the south was at a height of 0.879 m OD Malin Head, a difference of 0.22 m. Levelling established a difference of 0.15 m, thus, we assume the dGPS elevations had a vertical accuracy of 0.07 m.

### Statistical analysis

Elevation change readings from the SET stations and surface sediment organic matter contents north and south of the causeway were first tested for normality using a Kolmogorov–Smirnov test at a significance level of $p \leq 0.05$ and then evaluated using either a parametric *t*-test or a non-parametric Mann–Whitney *U*-test (in the case of organic matter contents, where independent samples could be assumed) or Wilcoxon Signed-Rank test (in the case of related water levels either side of the causeway), depending on whether samples met the normality/independence criterion.

## Results

### Marsh surface elevation change and accretion

Mean surface elevation change across all SET pin locations varied between the three measured time periods, from +0.1 mm (−3.9 to +2.5 mm) over the 9 months between the 19 November 2021 and 10 August 22 to 6.7 mm (+3.3 and +11.2 mm) over the following 12 months (10 August 2022 and 23 August 2023), to only 0.6 mm (−1.6 and +3.7 mm) between the 23 August 2023 and the 16 of August 2024 (another 12-month period).

Table 1 separates out surface elevation change measured at the two mature marsh sites either side of the causeway (for locations, see Figure 1), and Figure 2 shows cumulative elevation change and accretion for each of the four bar positions of the northern (SET 1, Figure 2a) and southern (SET 2, Figure 2b) SET locations.

At all but four SET bar positions, accretion exceeded elevation change on all the dates of measurement. The four exceptions all occurred over the period 10 August 2022 to 23 August 2023 at SET 1 (north of the causeway) at the 'east' and 'south' bar positions (where cumulative elevation change exceeded accretion by a factor of 1.1 at both positions) and at SET 2 (south of the causeway) 'north' and 'east' bar positions (where cumulative elevation change exceeded accretion by a factor of 3.0 and 1.2, respectively).

The greatest difference between cumulative accretion and elevation change over the full monitoring period (i.e., from 19 November 2021 to 16 August 2024) was observed at SET 2 (south of the causeway), where all bar positions showed accretion to exceed elevation change by at least a factor of 1.3 and by a maximum of 5.1 at the 'west' bar position (accretion of 8.9 mm compared to an elevation change of only 1.7 mm over the 33-month period) (see Figure 2b). Expressed as an annual rate, these figures translate to 3.2 mm year$^{-1}$ accretion and 0.6 mm year$^{-1}$ respectively.

Standard deviations in cumulative elevation change measurements were notably lower at SET 1 north of the causeway (1.9, 2.7 and 3.8 mm averaged for all four bar positions ($n = 36$ pin positions) for the three time periods, respectively) compared to SET 2 south of the causeway (6.2, 6.0 and 6.4 mm). This difference in the spatial variability at each of the SET locations is reflected in Table 1 and Figure 2.

Distributions of the $n = 36$ measurements at SET 1 or 2 only passed the Kolmogorov–Smirnov test for normality at a significance level of $p \leq 0.05$ for the second time period (10 August 2022–23 August 2023) at SET 1 (north of causeway) and the third time period (23 August 2023–16 August 2024) at SET 2 (south of the causeway). The non-parametric Mann–Whittney U-test (independent samples) confirmed that the elevation change over the full 33-month measurement period (19 November 2021–16 August 2024) was statistically significantly different ($p < 0.02$) between SET 1 (8.7 ± 2.7 mm) and SET 2 (6.3 ± 6.0 mm), north and south of the causeway, respectively. These rates translate to an annual elevation change of 3.2 mm year$^{-1}$ and 2.3 mm year$^{-1}$, respectively north and south of the causeway.

### Surface sediment characteristics

Organic matter contents within the surface sediments within 20 m of the SET stations ranged from 17.6% to 29.2% (mean of 21.4%) near the northern SET (SET1) and from 8.7% to 45.2% (mean of 30.8%) around the southern SET (SET2). While the range of organic matter content at the southern site exceeded that at the northern site, the southern site thus contained, on average, a higher organic matter content, and this difference was statistically significant (two-tailed t-test, $p < 0.05$). The data is summarised in Figure 3a.

Particle size distributions of surface sediments are shown in Figure 3b and indicate well-sorted, unimodal, fine sand. Mean particle size on the northern marsh was 149 μm, compared to 146 μm on the southern marsh, with sorting coefficients of 50.2 and 47.1 μm, respectively. Distributions were positively skewed (geometric skewness of 4.6 and 4.7 μm, respectively) and mesokurtic (kurtosis of 30.2 and 38.7 μm, respectively) both north and south of the causeway.

### Tidal water level fluctuations

Tidal stage curves for each of the six monitored tides are shown in Figure 4 within the context of tidal water levels recorded at Dublin tide gauge for the period 15 August 2023 to 15 September 2023. Tidal high water levels reached between 2.0–2.4 and 1.5–2.0 m ODM North and South of the causeway, respectively over the five tides, with the highest tides being those of 1 and 2 September 2023. While there were no marked differences in the timing of high water between the Dublin tide gauge and the two water level sensors north and south of the causeway, the height of the water level at the point of high water differed by between 0.05 m (tide of 31 August) and 0.45 m (tide of 1 September), with the water level highest in the southern and lowest in the northern lagoon.

**Table 1.** Mean cumulative elevation change (mm) from 19 November 2021 to 16 August 2024 and elevation change (mm) and accretion (mm) per time period (±1 standard deviation) at each of the two SET stations

| | Month | 0 | 9 | 21 | 33 |
|---|---|---|---|---|---|
| | Date | 19-Nov-21 | 10-Aug-22 | 23-Aug-23 | 16-Aug-24 |
| SET 1 (North) | Cumulative elevation change | 0 | 0.4 ± 1.5 | 7.6 ± 2.1 | 8.7 ± 2.7 |
| | Over previous: | | 9 months | 12 months | 12 months |
| | Elevation change | 0 | 0.4 ± 1.5 | 7.2 ± 2.4 | 1.1 ± 3.2 |
| | Accretion | 0 | 2.8 ± 1.0 | 5.1 ± 1.0 | 2.7 ± 1.8 |
| SET2 (South) | Cumulative elevation change | 0 | −0.2 ± 4.8 | 6.1 ± 5.8 | 6.3 ± 6.0 |
| | Over previous: | | 9 months | 12 months | 12 months |
| | Elevation change | 0 | −0.2 ± 4.8 | 6.2 ± 4.3 | 0.2 ± 4.9 |
| | Accretion | 0 | 2.9 ± 1.0 | 2.4 ± 1.3 | 5.6 ± 2.8 |

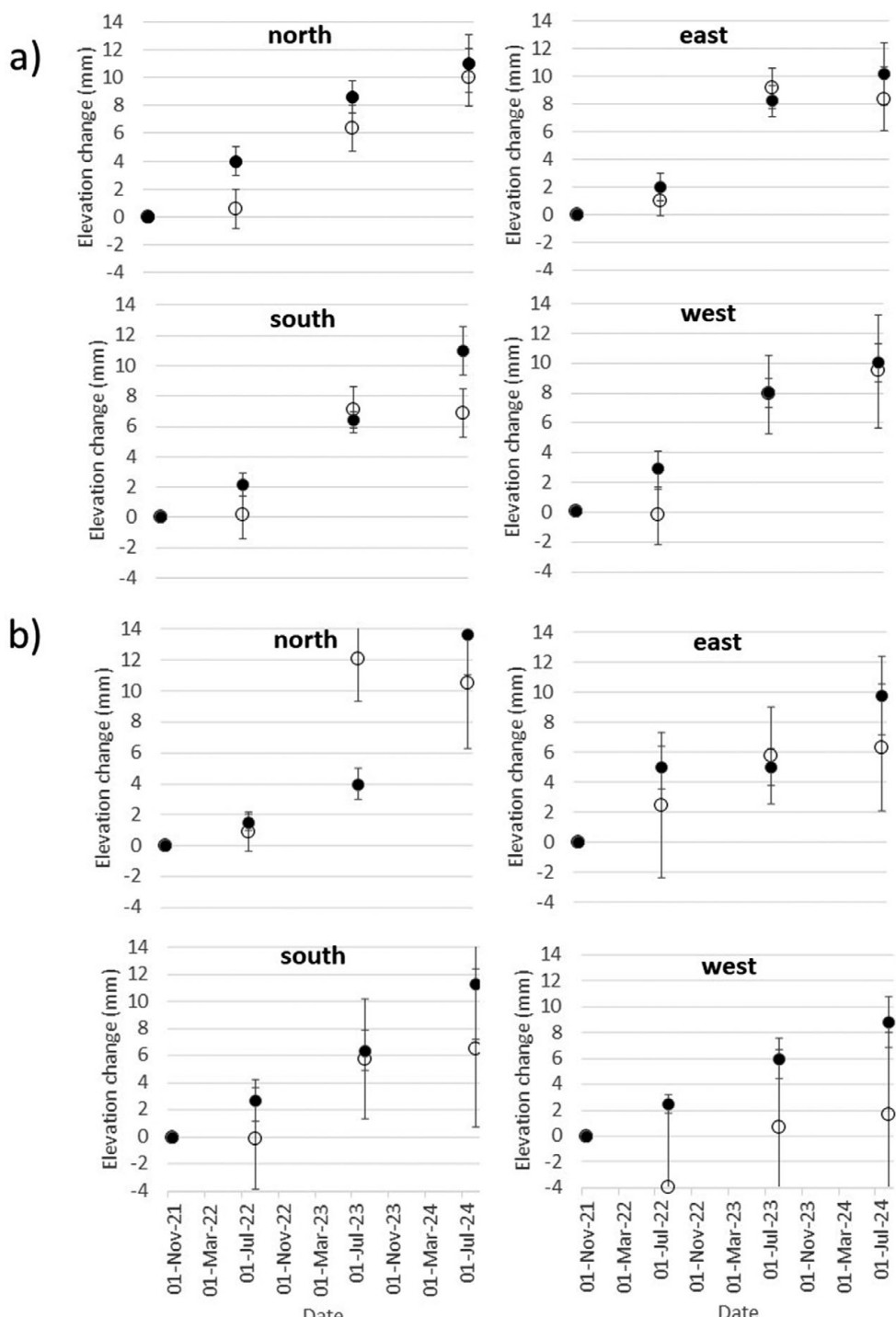

**Figure 2.** Cumulative elevation change (open circles) and accretion (black dots) for each of the four SET bar orientations at (a) SET 1 north of the causeway and (b) SET 2 south of the causeway (*note*: error bars show ±1 standard deviation).

With the exception of the tide of 31 August, high water levels in the southern lagoon exceeded those recorded at Dublin's tide gauge by between 0.13 and 0.15 m, and those recorded at Dublin tide gauge exceeded those in the northern lagoon by up to 0.33 m (3 September tide). Thus, differences in the maximum tidal water level between the northern and southern lagoons varied from 0.45 to 0.47 m on the five monitored spring tides between the 1 and 5 September 2023.

To rule out any potential instrument malfunctions that might explain the different patterns we observed between the flood pattern on the tide of the 31 August and those of 1–5 September (Figure 4), we decided to (a) provide contextual data on meteorological conditions from Dublin Airport's weather station and (b) record water levels on one further tide on 22 September 2023. The results of this additional analysis are presented in Figure 5. The tide of 22 September 2023 was a much lower tide (reaching a water

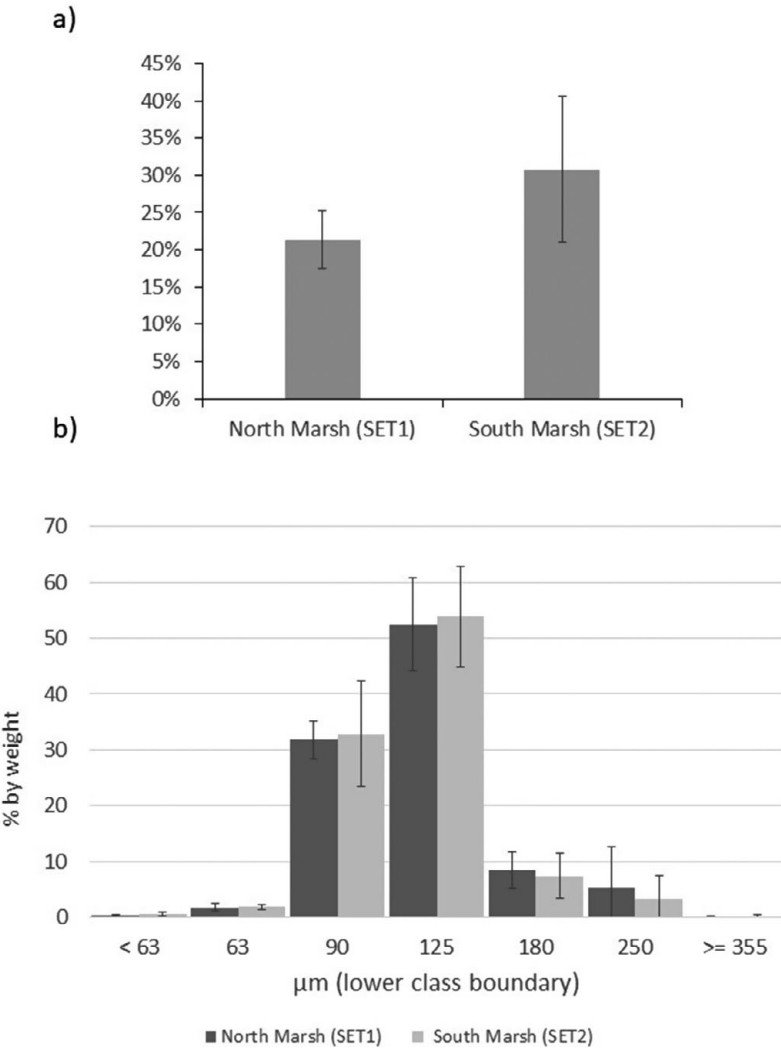

**Figure 3.** Percentage organic matter content (a) and particle size distributions (b) on the northern and southern marsh sites near the two SET stations.

elevation of only 1.34 and 1.23 m OD (Malin) at the Dublin tide gauge and north and south of the causeway) and, similar to the tide of 31 August, showed almost identical flooding/draining patterns of the upper tidal flats/salt marshes above 1 m OD (Malin). The differences observed on the five tides of 1 through to 5 September (Figure 4) are thus indicative of a real pattern.

In the flood phase of the tide, the 1 m OD (Malin) elevation was exceeded first at the Dublin tide gauge, then in the southern lagoon (ca. 15–25 min later) and then (a further 10 min later) in the northern lagoon. In the ebb phase, the water level remained above 1 m OD (Malin) elevation for around 30 min longer in the southern lagoon than at Dublin tide gauge, with the northern lagoon water levels falling below 1 m OD (Malin) elevation between 5 min (31 August tide) and up to 25 min (5 September tide) after they did so at Dublin tide gauge. These differences in timing resulted in the duration of tidal flooding above the 1 m OD (Malin) contour varying between the Dublin tide gauge (255–305 min), the south lagoon sensor (190–260 min), and the north lagoon sensor (280–320 min). These inundation durations above 1 m OD (Malin) elevation were highly statistically significantly different between the Dublin tide gauge and the northern lagoon sensor location ($p < 0.03$) and between the southern lagoon and northern lagoon

sensor ($p < 0.03$) but less so between the Dublin tide gauge and the southern lagoon ($p < 0.09$) (Wilcoxon Signed Rank Test).

## Discussion

Given their adaptation to salinity stress and lower oxygen availability, salt marsh vegetation communities rely on the surfaces they grow on to sustain their elevations relative to sea level (Allen, 2000). As sea level rises, this can only be achieved through a combination of sufficient surface accumulation of biological matter, below-ground root contributions to soil volumes, and tidally imported organic and inorganic sediment (French, 2006; Saintilan et al., 2022). Compaction and decomposition of matter over time (years to decades) means that, in most cases, elevation change over multi-annual time scales is less than the thickness of the annually deposited layers, that is, measurements obtained from using SET-type methods are often lower than those obtained by accretion methods (e.g., marker horizons) (Cahoon et al., 2006).

Our study is the first to combine accretion and elevation change measurement on an Irish salt marsh. In this case of the back barrier lagoon salt marshes of Dublin Bay, neither the marshes north

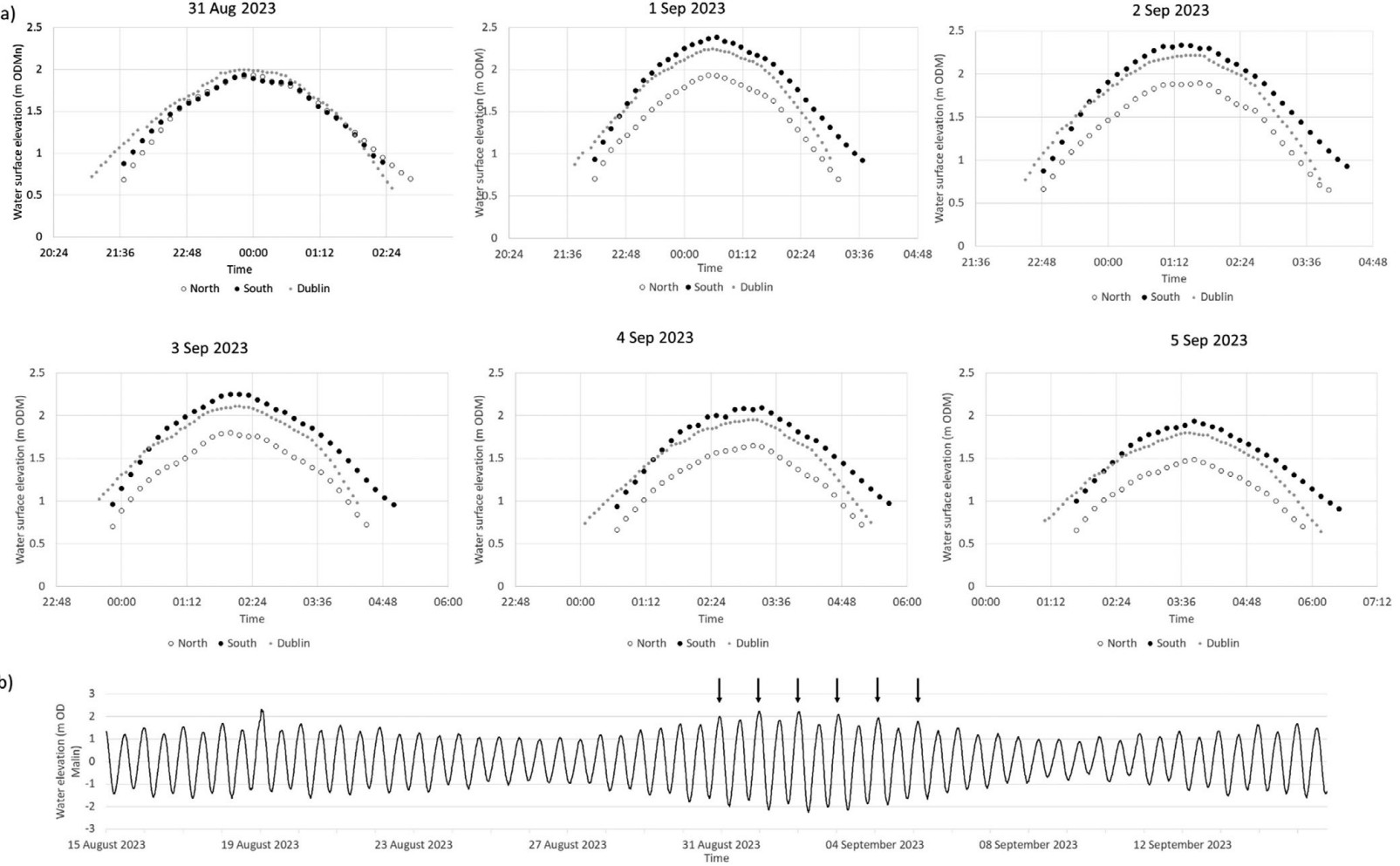

**Figure 4.** Tidal stage curves (a) for the six observed largest spring tides (31 August to 5 September 2023) north and south of the causeway as well as at the Dublin tide gauge and (b) tidal water level fluctuations at Dublin tide gauge between 15 August and 15 September 2023, placing the six observed tides (arrows) in context.

(elevation increase of 3.2 mm year$^{-1}$ at SET 1) nor those south of the causeway (elevation increase of 2.3 mm year$^{-1}$ at SET 2) are currently experiencing a surface elevation rise that matches the latest estimates of recent rates of sea-level rise (7 mm year$^{-1}$ over the period 1997–2016; Nejad et al., 2022). Over our measurement period, accretion (10.6 (SET 1) to 10.9 (SET 2) mm year$^{-1}$) out-paced elevation change (8.7 (SET 1) to 6.3 (SET 2) mm year$^{-1}$), suggesting mean shallow subsidence ('autocompaction'; Cahoon et al., 2006) rates of 0.7 and 1.7 mm year$^{-1}$ north and south of the causeway, respectively, over the past 33 months. These rates are low compared to marshes on the east coast of the UK (e.g., 3–4 mm year$^{-1}$ in North Norfolk, UK (Cahoon et al., 2000; Spencer et al., 2012) and 3–10 mm year$^{-1}$ in Essex (Schuerch et al., 2019), and the lower shallow subsidence rates at the northern marsh site (SET 1) suggest that added sediment is converted more efficiently into a change in surface elevation than at the southern marsh site (SET 2).

Our sediment organic matter measurements bear out this explanation. Although we did not find any statistically significant difference in the particle size distributions, surface sediment scrapes collected to the north of the causeway contained, on average, statistically significantly less organic matter (21.4%) compared to those south of the causeway (30.8%). Our results correspond closely to the mean organic matter contents of 22.5% and 19%–24% reported by Grey et al. (2021) and Doyle and Otte (1997) for the salt marshes of North Bull Island. As has been observed in a global analysis of salt marsh response to relative sea level rise by Saintilan et al. (2022), higher organic matter contents (as we record on the southern marsh) tend to lead to greater autocompaction but may also reasonably explain the higher variability in elevation change measurements here (due to spatially varied rates of litter, below-ground root production, and resulting microtopographic variability) compared to the northern marsh site.

At the north arm position of the southern (SET 2) site, elevation change temporarily exceeded accretion over the 2022–2023 measurement period (see Figure 2). The reason for such localised variations in elevation versus accretion change measurements likely reflects the effects of seasonal and patchy deposits of fine filamentous and spatially and temporally non-uniform layers of organic material on the surface. Such temporal and spatial variability in the measurements underscores the benefit of investing in several field monitoring stations maintained and measured over decades.

The 0.07 m difference in surface elevation at the SET locations (1.29 m and 1.36 m OD Malin, SET 1 and SET 2, respectively; Figure 1b) might explain the difference in mean elevation change (8.7 (SET 1) to 6.3 (SET 2) mm year$^{-1}$ north and south of the causeway, respectively) based on the premise that a greater hydro-period results from lower elevations in the tidal frame (French, 2006). However, our empirical evidence for the flooding and draining patterns north and south of the causeway in the back-barrier lagoon helps to elucidate the present-day sedimentary environments in the two separate (north and south of the causeway) lagoons in greater detail. With the exception of our first monitored tide on 31 August (when the north and south lagoon water levels tracked similar elevations and both closely tracked water levels at Dublin's tide gauge (Figure 4)), we observed a consistently different tidal regime between the Dublin tide gauge (first to flood with second highest maximum water levels), the southern lagoon (second to flood with the overall highest maximum water levels), and the northern lagoon (last to flood with the lowest maximum water levels) (Figure 4; Figure 1 for locations). Meteorological conditions are likely to play an important part in influencing the speed and timing of flooding/draining. The reasons for these differences are not immediately clear, as wind conditions during the tidal flooding varied considerably in both wind speed and direction, and both the tides of 3 September and 22 September were characterised by a strong westerly (offshore) wind (Figure 5) but resulted in very different inundation patterns (Figure 4). According to the Irish meteorological institute's (Met Éireann's) weather observation website (wow.met.ie), the meteorological station at Howth, a short distance (<2 km) from North Bull Island, recorded a rise in pressure from 1,009 mb during the first tide (31 August) to 1,026 mb on the 3 September tide and then fell to 1,019 mb on the 5 September tide and 1,000 mb on the tide of 22 September, with no consistent influence on the inundation patterns observed (Figure 4). A further investigation into the flooding and draining patterns observed in this study is now needed, and a fine-scale numerical modelling approach as well as a longer empirical dataset of water level fluctuations may shed further light on the dynamics of tidal flooding into the north and south lagoons.

### *Implications for salt marsh resilience to sea level rise*

In the context of sediment deposition on intertidal and particularly salt marsh surfaces, both inundation depth (i.e., water level relative to bed level) and inundation duration, together referred to as 'hydroperiod', play a key role in determining the opportunity for sediment deposition (French, 2006). In the case of North Bull Island, the statistically higher maximum tidal water levels (0.45–0.47 m higher in the southern lagoon compared to the northern lagoon) appear to be compensated by the longer flood duration in the northern lagoon (280–320 min compared to 190–260 min in the southern lagoon (above the 1 m OD (Malin) elevation contour)).

The statistically significantly higher elevation change rates recorded at the northern lagoon SET site (and arguably the lower variability in these measurements), along with the significantly lower organic matter contents in the surface sediments in the northern lagoon, suggest that the longer duration of flooding in the northern lagoon is able to facilitate a more positive response of the marsh surface to flood frequency and duration. Thus, while our results indicate that neither of the two lagoon environments show a sufficient marsh surface elevation rise to keep pace with recently observed longer-term (decadal) local rates of sea level rise, we postulate here that it is the greater relative contribution of minerogenic sediment that may currently allow the northern lagoon's marsh surface to more efficiently convert deposited sediment into elevation change through lower shallow subsidence ('autocompaction') rates as has been observed elsewhere (Saintilan et al., 2022).

The source of the allochthonous, minerogenic sediment at the dominant size of 146–149 μm (2.9 phi) recorded in our marsh surface scrapes is likely a combination of resuspended sediment from the outer parts of Dublin Bay, which is reportedly dominated by grain sizes in the fine sand fraction <3 phi (Harris, 1980). Harris (1977) also reported well-sorted fine sands with a dominant grain size range of 124–250 μm for Dollymount strand on the seaward side of the island. However, over recent years (2005–2021), dune erosion (a landward retreat of the dune vegetation line in excess of 10 m) has been observed in the far northern reaches of the island and close to Sutton Creek, the channel through which the tide enters the northern lagoon (Mathew et al., 2019; McClung, 2021). It is possible that this eroded material is transported into the lagoon and deposited as part of the relatively higher minerogenic content we observed in the northern lagoon marshes. In this context, it is important to also note the difference in the cross-shore profile from

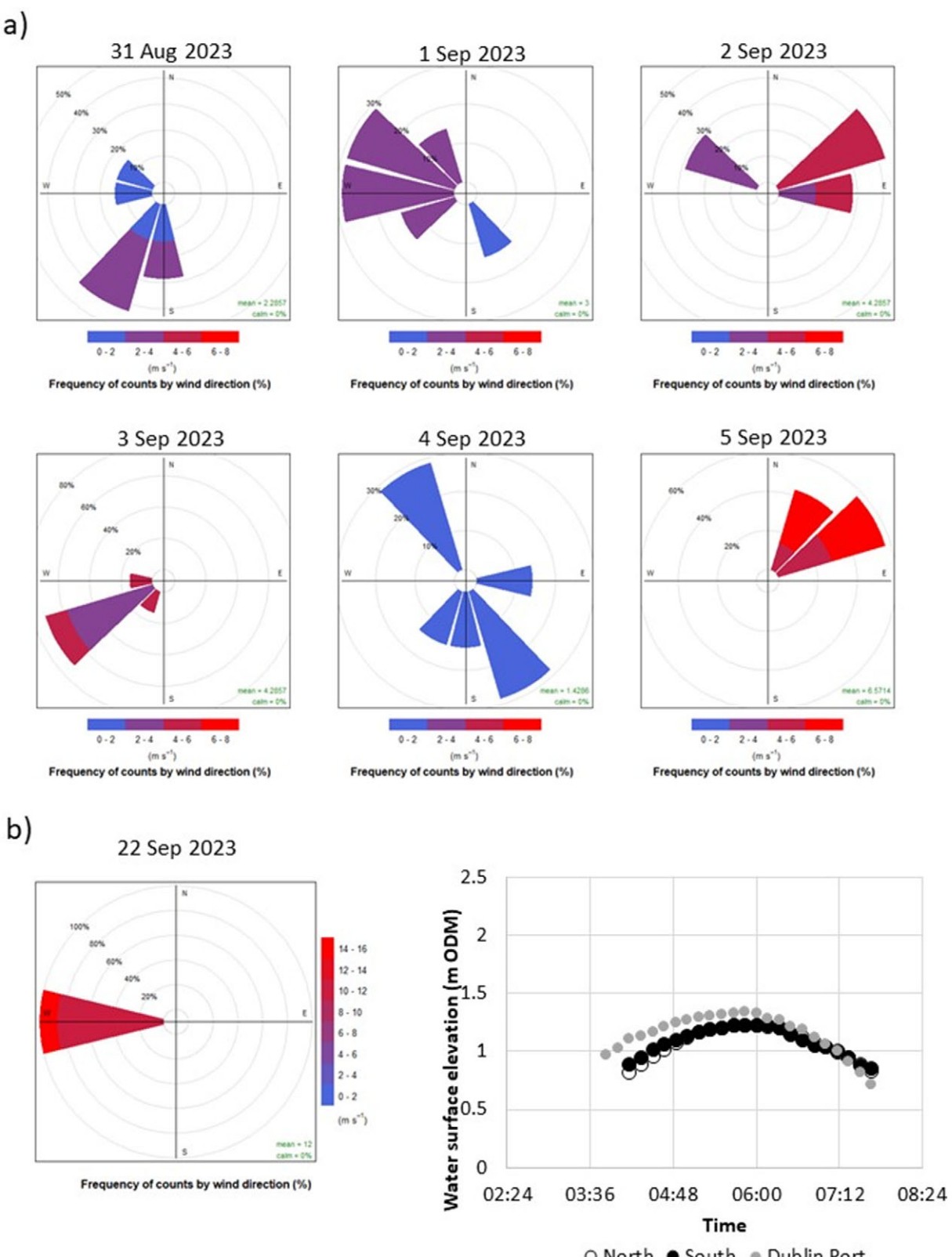

**Figure 5.** Wind direction and speed counts (10-min intervals) for (a) the duration of tidal inundations displayed in Figure 4 and (b) the inundation during the 22 September tide (left) for which water levels were also monitored (right).

the tidal flat onto and across the salt marsh surface towards the dunes of the barrier island (Figure 1b). The northern profile retains the convex shape typical of tidal marsh to mudflat transitions in an earlier stage of morphological evolution, while the more concave marsh surface in the south suggests a more mature marsh (French and Stoddart, 1992).

Further research now needs to continue to build on the dataset reported here and supplement it with a spatially explicit model of water and sediment fluxes in different meteorological as well as tidal conditions.

## Conclusion

Many coastal salt marshes are now protected through nature conservation legislation frameworks, and access to those types of coastal spaces plays an important role in rebuilding human-nature connections, particularly in urban settings. Our study on the impact of a constructed access causeway on patterns of accretion, elevation change, and observed tidal inundation dynamics of the now split back-barrier lagoon suggests that the causeway has significantly affected the natural functioning of the lagoon and its salt marsh habitats. This highlights, in particular:

1) the delicate balance arising between the two related aims of making the most conservation legislation-protected urban coastal space in Ireland accessible while, at the same time, facilitating the future persistence of the landforms and ecosystems that make this space ecologically and societally valuable; and
2) the need for informed adaptation planning that is based on a sound understanding of the very processes that ensure the sustainability of valued community resources, particularly when those resources are shaped, formed and maintained by a suite of complex biophysical dynamics.

Given the necessity to safeguard valuable salt marsh ecosystem services in the context of widespread pressures of human development in the coastal zone, it is imperative that knowledge is built around the impact of such developments on the future resilience of salt marsh. This is particularly important when it comes to the impact of an accelerated rise in sea level and the dependence of salt marsh surfaces on adequate sediment deposition.

**Open peer review.** To view the open peer review materials for this article, please visit http://doi.org/10.1017/cft.2025.6.

**Data availability statement.** Data are available on request from the authors

**Acknowledgements.** We are also grateful to Dublin City Council (particularly Patrick Corrigan) and the National Parks and Wildlife Service for supporting our research on North Bull Island, as well as to Dr Elaine Treacy and Dr James Canavan of the technical staff at Trinity College for their invaluable assistance and support around the field and laboratory work carried out for this study, as well as many members of Trinity's Coastal Research Group (particularly Davide Tognin, Lei Chen and Jenny Clarke) and students of the 'Coastal Wetlands' module, who assisted with field and laboratory work.

**Author contribution.** The lead author (Möller) wrote the draft and final version of the paper with significant input from the co-author (O'Leary), who was instrumental in assisting with the implementation, design and execution of the deployment, data capture and analysis.

**Financial support.** We are grateful to the Trinity College E3 initiative for providing the funding for K O'Leary's research assistantship and to the European Union REWRITE ('REWilding and Restoration of InterTidal sediment Ecosystems for carbon sequestration, climate adaptation and biodiversity support') project (grant agreement 101081357) for facilitating the data analysis and writing of the manuscript. Views and opinions expressed are however those of the author(s) only and do not necessarily reflect those of the E3 initiative or the European Union. Neither the European Union nor the E3 initiative can be held responsible for them.

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
