## [Reviewer Report]

This is an interesting location due to its geomoprhological history and current status as a biosphere reserve. The paper discusses measurements made in the marsh on the island and the tidal dynamics of the area. I have a number of comments on the details of the paper below. But in summary I feel this would better highlight the potential conflict between access/use and sustainability with more context on the past development of the system and how the natural elemanets, especially the marshes, are currently valued by society. This can then set the field study in context andall the authors to better draw the isights they get from their data to the larger issues surrounding the site, that are alluded to at the start. The paper need not be longer (I have a number of suggestions on how methods and discussion of data details can be shortened) but can better then address the title ' balancing conservation and human use/value.

Detailed comments and suggestions below:

Page 3, line12. I am not seeing the connection between these two ideas. Nature based solutions must provide a particular type of ‘service’ that is not well explored in the early in the ES literature, including that cited. I think what is missing in this text is the problem that the solution is designed to address. Assuming that it is to ‘break, divert, or reflect the sea’s energy’ then I think those ideas emerge much later than Costanza. The 2004 Indian Ocean tsunami drew a lot of attention (appropriately or not).

Page 3 line 29. I think it would be really useful to step through this in a historical context. For those not familiar with the area ‘what came first’ and which action had which consequences is tough to tease out. A sequence of maps, either historical or conceptual’ would hep the reader understand this complex evolution.

Page 3, line 40. I suggest you reframe this as a goal of the paper unless you really are predicting the future. Also - you seem to be treating access as a binary - it either exists in its current form or it doesn’t. There may be other modes of access that could be explored to support the sustainability that may be in question. This may be a story more about adaptation to enable the system (human and natural) to cope with future conditions.

Page 3, line 52. As noted above, I think the reader needs a better understanding of the system before the jetties. The next paragraph indicates there were already shallow areas - but no island. More detail please.

Page 4, line 3-4. Unclear what is meant by ‘ever more complex and mature inter-connected bio-physical system’

Page 4, line 5. This implies there was access before the causeway - how? See previous comment on exploration of the nature of access.

Page 4, line 12. The causeway is not marked well on Figure 1. The directions seem to be NE and SW but the figure is not clear.

Page 4, line 14. What is the date/source of the vegetation information?

Page 4, line 17. What period/dates are the SLR rates for?

Page 4, line 20. It would be useful to clarify if the causeway is a complete blockage or whether is any exchange. Also - is there exchange through the breakwater?

Page 4, lines 36-45. Rather than all this detail I suggest you rely on the citation and focus more on what the measurements tell us (surface elevation change relative to an assumed fixed subsurface point). Seems very odd to have all this detail on the measurement but so little detailed context on the site.

Page 4, lines 53-60. This detail can also be handled through a citation and focus the text on distinguishing what these measurements tell us (what is happening on the surface of the marsh)

Page 4, line 9. See previous comments on the level of detail, use of citations and more focus on why these measurements are helpful.

Page 6, lines 3-11. The text seems to just repeat information which could easily be in the figures or the tables

Page 6, line 40. Statistical approaches should be described under methods.

Page 8, line 26. How do you know this is deposited sediment? There are so many processes going on here its is unclear what evidence you have to key in on just one.

Page 8, line 38. This is quite a major assumption about greater autocompaction as you don’t have any indication of the type of organic matter. Bulk density data would be useful to understand/infer compactional processes

Page 8, line 39-57. I am not sure it is worth trying to understand the local; variation in processes around the SET. The study is premised on the use of 2 RSETs that effectively characterize the marsh conditions on either side of the causeway. To analyze these data at this level without a detailed description of how local conditions, like microtopography or vegetation distribution, occur around the SEWTs just raises more questions than it answers.

Page 9, line 50-54. While this is likely the case you don’t have direct evidence

Page10, line 40. A guarantee seems like a lofty and unachievable aim in a dynamic coastal setting

Page 10, line 42. I don’t think you have made a case for more monitoring. Rather I think your final point might be more about adaptation planning that is based on understanding of those processes that ensure the sustainability of the resources that the community values.

---

## [Reviewer Report]

General comments

The manuscript deals with a very relevant topic. Very nice to see what the impact of man-made constructions on the sediment dynamics is in an urbanized region. There are not a lot of studies that highlight this interesting topic.

The aim of the study is well described at the end of the introduction.

The methods are clearly described, and the data are properly analyzed. The statistics are solid and all figures showing the results of detailed measurements do have proper uncertainty estimations (figures 2 and 3). The figures are also of high quality. The location of the study sites and the detailed measurement points are clearly presented, and the hydrodynamic conditions (water levels in figure 4) are easy to interpret.

The authors are also critical about their own findings, and they do quite a lot of additional analysis to exclude some of their hypotheses. I really liked the additional analysis around figure 5 (page 7, lines 29-40).

The discussion was very nice. They included many other studies and reflected on similarities and differences with their observations. However, the reflection on the detailed measurements of sedimentation rates uses mainly studies from the UK and Ireland.

The paper is well structured and well written. There is a nice flow and a very good coupling between the text and the figures.

Enjoyed reading it!

Detailed remarks:

Page 2, lines 49-51 (2, 49-51): sentence is a bit complicated.

3, 54: typo (Kennedy, 1949)

4, 19: typo et al., 2022 (comma lacking)

5, 16: in keeping with Grey et al. ??

5, 35-44: what is the frequency of the measurements, what is the burst duration and what is the burst interval? Do you present average water levels over a distinct interval? (is what I interpret from figure 4).

6, 14: Sometimes you write Figure with a capital, often without. Good idea to write it all with or without.

9, 24: typo 2019 mb should be 1019 mb

9, 24: can expect that the barometric effect will have an effect on the water levels. Normally ca. 1 mb = 1 cm.

10, 20-47: Conclusions. This is more than only conclusions. You start with a relevance of the topic (lines 23-30) and present your conclusion in line 31-34. Thereafter, you place it in a societal context.

---

## [Editor Report]

There is one recommendation of minor revision and one of major revision here. On reading the submission alongside the reviewers' comments, I do think that a major revision is needed.

---

## [Reviewer Report]

Dear authors,

Just went through the resubmitted manuscript and the comments to the reviewer.

You addressed almost all comments from both reviewers, and I was happy with the changes made to the first manuscript. These changes were mainly focused on the last part of the introduction, the field site description, and the methods. The local conditions and the history of the site are even better described now. Figure 1 is also easy to read with all additional information in the text.

The method description of the topography changes is more precise. The aim of this study is well defined, and the discussion is clear.

I have no additional comments to the resubmitted manuscript.

Best regards.

---

## [Reviewer Report]

Thank you for addressing my comments. I have two very minor things to point out on the current version

- first line under Methods mentions three key questions. I had trouble locating these. Either rephrase or make them more prominent in the previous text.

- Under results ‘At all but four SET bar positions, accretion exceeded elevation’ - should be accretion exceeded ‘elevation change’.